# RELEASE-HF study: a protocol for an observational, registry-based study on the effectiveness of telemedicine in heart failure in the Netherlands

Jorna van Eijk [ORCID],[1] Kim Luijken,[2] Tiny Jaarsma,[1] Johannes B Reitsma,[2] Ewoud Schuit,[2] Geert W J Frederix,[2] Lineke Derks,[3] Jeroen Schaap,[4,5] Frans H Rutten [ORCID],[1] Jasper Brugts,[6] Rudolf A de Boer,[6] Folkert W Asselbergs,[7,8] Jaap C A Trappenburg,[9] RELEASE-HF Investigators

FWA and JCAT contributed equally.

For numbered affiliations see end of article.

**Correspondence to**
Jorna van Eijk;
j.vaneijk-4@umcutrecht.nl

## ABSTRACT

**Introduction** Meta-analyses show postive effects of telemedicine in heart failure (HF) management on hospitalisation, mortality and costs. However, these effects are heterogeneous due to variation in the included HF population, the telemedicine components and the quality of the comparator usual care. Still, telemedicine is gaining acceptance in HF management. The current nationwide study aims to identify (1) in which subgroup(s) of patients with HF telemedicine is (cost-)effective and (2) which components of telemedicine are most (cost-) effective.

**Methods and analysis** The RELEASE-HF ('REsponsible roLl-out of E-heAlth through Systematic Evaluation – Heart Failure') study is a multicentre, observational, registry-based cohort study that plans to enrol 6480 patients with HF using data from the HF registry facilitated by the Netherlands Heart Registration. Collected data include patient characteristics, treatment information and clinical outcomes, and are measured at HF diagnosis and at 6 and 12 months afterwards. The components of telemedicine are described at the hospital level based on closed-ended interviews with clinicians and at the patient level based on additional data extracted from electronic health records and telemedicine-generated data. The costs of telemedicine are calculated using registration data and interviews with clinicians and finance department staff. To overcome missing data, additional national databases will be linked to the HF registry if feasible. Heterogeneity of the effects of offering telemedicine compared with not offering on days alive without unplanned hospitalisations in 1 year is assessed across predefined patient characteristics using exploratory stratified analyses. The effects of telemedicine components are assessed by fitting separate models for component contrasts.

**Ethics and dissemination** The study has been approved by the Medical Ethics Committee 2021 of the University Medical Center Utrecht (the Netherlands). Results will be published in peer-reviewed journals and presented at (inter)national conferences. Effective telemedicine scenarios will be proposed among hospitals throughout the country and abroad, if applicable and feasible.

**Trial registration number** NCT05654961.

## STRENGTHS AND LIMITATIONS OF THIS STUDY

⇒ A strength of this study is the use of nationwide routine clinical care data linked to national registry databases to capture current data on telemedicine use, the characteristics of patients with heart failure, treatments and clinically relevant outcomes with a 1-year follow-up.

⇒ Another strength is that data from the heart failure registry in the Netherlands and a separate data collection on telemedicine features using additional questionnaires allow consideration of heterogeneous treatment effects across the heart failure population and telemedicine components.

⇒ A limitation is that most data on telemedicine characteristics will be collected at the hospital level, rather than at the individual patient level.

⇒ Another limitation is that the cost-effectivity analyses are specific to the Dutch healthcare system, which may hamper generalisability of findings to other settings or countries.

⇒ A final limitation is that selection bias can occur when healthcare providers do not include all patients in the outpatient clinic in the heart failure registry because the registry is not incorporated in care pathways in electronic health record systems.

## INTRODUCTION

Heart failure (HF) poses a major socio-economic and patient burden, and healthcare systems are seeking innovative health models to support care. Optimised guideline-directed medical therapy, self-care (ie, healthy diet, medication adherence, exercise) and adequate monitoring of vital signs and symptoms may all help reduce HF-related morbidity.[1–3] Health models therefore use telemedicine as a tool to support patients in optimising HF management, self-care support and symptom monitoring to improve care and prevent (re)hospitalisation.[2–4]

'Telemedicine' is a heterogeneous intervention that includes a wide range of digital technologies (smartphones, mobile wireless devices/sensors, video connections, implantable devices, etc) exchanging digital health information between patients and clinicians to support and optimise the care process remotely.[5 6]

Many meta-analyses have evaluated the (cost-)effectiveness of telemedicine in patients with HF.[7–19] Overall, these meta-analyses point towards a positive effect of telemedicine on hospital admission, length of hospital stay, mortality and costs.[2 7–19] However, the effects are heterogeneous across subgroups[20–24] and vary with telemedicine components, such as type of monitored functions, number of alerts and risk of alert fatigue, contact with healthcare professional, type of device, and different quality of usual care.[7 14 15 19 25] This can lead to problems in comparability of studies. Understanding of the cost-effectiveness of telemedicine is fragmented because studies do not evaluate the same costs, with the focus shifting between the purchase costs of telemedicine, personnel costs and variation in the costs of telemedicine components.[4 12 16]

As a result, HF guidelines lack specific recommendations on *how*, *when* and in *whom* telemedicine should be provided. Despite the lack of solid evidence and advice, payers (eg, procurement officers, insurance companies) and patient organisations are advocating to accelerate the implementation of telemedicine for patients with HF in outpatient clinics.[26] Consequently, telemedicine is implemented in different formats with varying objectives, intervention components and implementation strategies. To reveal the real potential of telemedicine for patients with HF and its implementation in everyday practice, clarity on *how*, *when* and in *whom* and which telemedicine components are (cost-)effective is essential.

Here, we present the protocol for the RELEASE-HF ('REsponsible roLl-out of E-heAlth through Systematic Evaluation – Heart Failure') study. In this study, we focus on both patient-related uncertainties in telemedicine use and telemedicine intervention-related uncertainties from a clinical as well as an economic perspective. Our specific study objectives are to examine (1) which patient characteristics are related to an increase in the number of days spent alive without unplanned hospitalisations within 1 year when telemedicine is part of HF care compared with regular HF care; (2) which components of telemedicine as part of HF care lead to an increased number of days alive without unplanned hospitalisations within 1 year; (3) which characteristics of patients with HF are related to cost-effectiveness when telemedicine is part of HF care compared with regular HF care; and (4) which components of telemedicine as part of HF care are cost-effective.

## METHODS AND ANALYSIS
### Study design
RELEASE-HF is a nationwide, observational, registry-based cohort study across multiple hospitals in the Netherlands.[27] The study collects routine data longitudinally.

We record the health status of and cardiac interventions in patients with HF and simultaneously observe the natural (de)implementation of telemedicine. RELEASE-HF is linked to the Heart4Data consortium which started in 2022 to create a national and sustainable infrastructure for cardiovascular registry-based research in the Netherlands.[28] This infrastructure will develop a framework for the governance, ethical, legal, financial, information technology and methodological factors necessary for registry-based research.

### Progress and time plan
RELEASE-HF started in June 2021 and will last 4 years (figure 1). This study includes a collaboration with at least 29 hospitals that signed a letter of commitment. Additional hospitals may enter the study later if they can complete a 12-month follow-up per included patient through the HF registry.

### Data sources
RELEASE-HF will combine various sources of data: the Netherlands Heart Registration (NHR) HF registry, other national databases, interviews, electronic health records (EHR) and telemedicine-generated data (figure 2). A detailed description of each data source is provided in the following:

#### HF registry
The HF registry is an ongoing quality registry in the Netherlands facilitated by the NHR, a non-profit organisation that aims to contribute to quality improvement and safety in cardiac care.[29 30] The HF registry serves as an ongoing learning healthcare system through benchmarking and quality control. Since 2019, it has been introduced as a voluntary, nationwide HF registration. Patient data are collected non-consecutively, and for the registry no informed consent is asked. Data of included patients are collected at three timepoints: at baseline, 6 months and 12 months from the time since diagnosis. The HF registry includes variables on three levels defined in a data dictionary: patient characteristics, treatment characteristics and clinically relevant outcomes. The data dictionary is a dynamic document (ie, the included variables and definitions may change over time) because it is based on current HF guidelines and compiled by a committee consisting of delegated cardiologists from Dutch hospitals. A detailed up-to-date data dictionary with an overview of the variables collected within the HF registry is always published on the NHR website. Online supplemental material 1 provides an overview of the data dictionary of the HF registry, September 2022 version; the content is subject to change.[31]

#### National registries and databases
If legally and technically feasible, we plan to link the HF registry data, specific to RELEASE-HF, to external data sources to (1) complement missing variables, (2) enrich and validate the HF registry, and (3) reduce registration burden for healthcare professionals/data managers. The

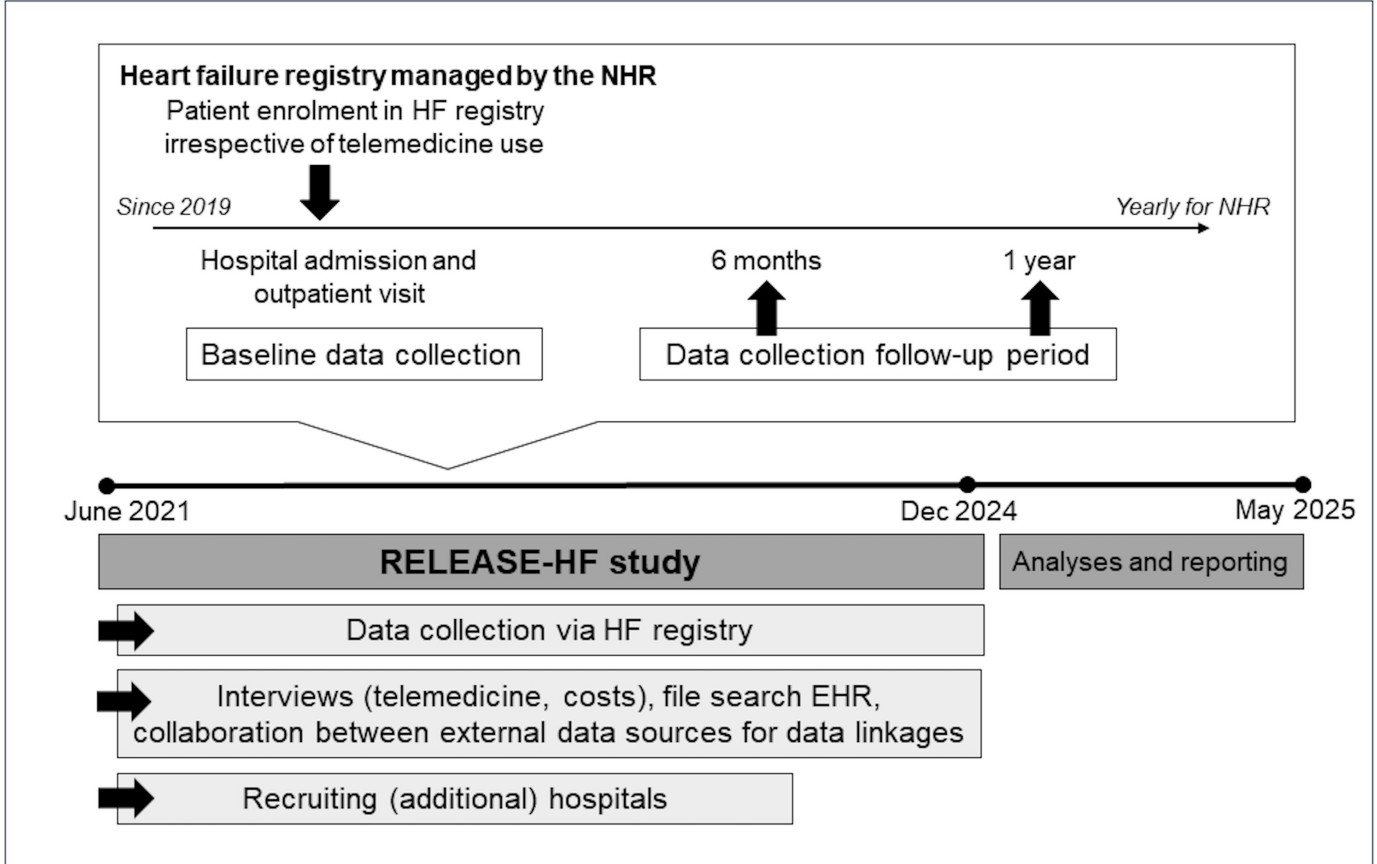

**Figure 1** Overview of the time plan for the RELEASE-HF study. EHR, electronic health record; HF, heart failure; NHR, Netherlands Heart Registration; RELEASE-HF, REsponsible roLl-out of E-heAlth through Systematic Evaluation – Heart Failure.

external data sources considered are Statistics Netherlands (CBS), the national health insurance database (VEKTIS), the national registry for hospital care (Dutch Hospital Data) and the national database for drug utilisation and drug safety (PHARMO). The variables of interest are visits to the outpatient clinic, to the emergency room and to the general practitioner related to HF, admission days at an intensive care unit, and treatment characteristics specific to HF (ie, medication as class of diuretics).

### Interviews on the use of telemedicine
Separate data collection will be performed using semi-structured interviews among clinicians and finance department staff on the features and costs of telemedicine at the hospital level. Online supplemental material 2 provides the interview guide.

### EHR and telemedicine-generated data
File search in the EHR of patients and telemedicine-generated data will be used to identify telemedicine components at the patient level.

### Study population
The HF registry comprises patients admitted to a Dutch hospital or to an outpatient clinic for HF irrespective of telemedicine use (figure 1). All patients who meet the diagnostic criteria for HF and its phenotype, according to the European Society of Cardiology (ESC) 2021 guidelines

on HF, are included.[1] If a patient has been diagnosed with HF in a setting other than the one where the patient currently presents (primary, secondary or tertiary care), the patient will also be included.[31] Current HF registry is not yet incorporated and implemented in care pathways in EHR systems; therefore, selected patients fulfilling the NHR criteria are included in the HF registry and not on a consecutive basis or full coverage of all patients at the outpatient HF clinic. This may lead to selection bias. To prevent this selection bias, a researcher of RELEASE-HF is alerting healthcare professionals regularly to include all patients with HF in the HF registry, regardless of their telemedicine use, treatment and disease severity. Additionally, national working groups are progressing to fully incorporate the HF registry into the EHR.

### Outcome measures
#### Primary outcome
The number of days alive without unplanned hospitalisation within 1 year of follow-up is derived from the number of unplanned hospital days as collected in the HF registry (table 1).

#### Secondary outcomes
The secondary outcomes are all-cause mortality, functional status, health status, health-related quality of life

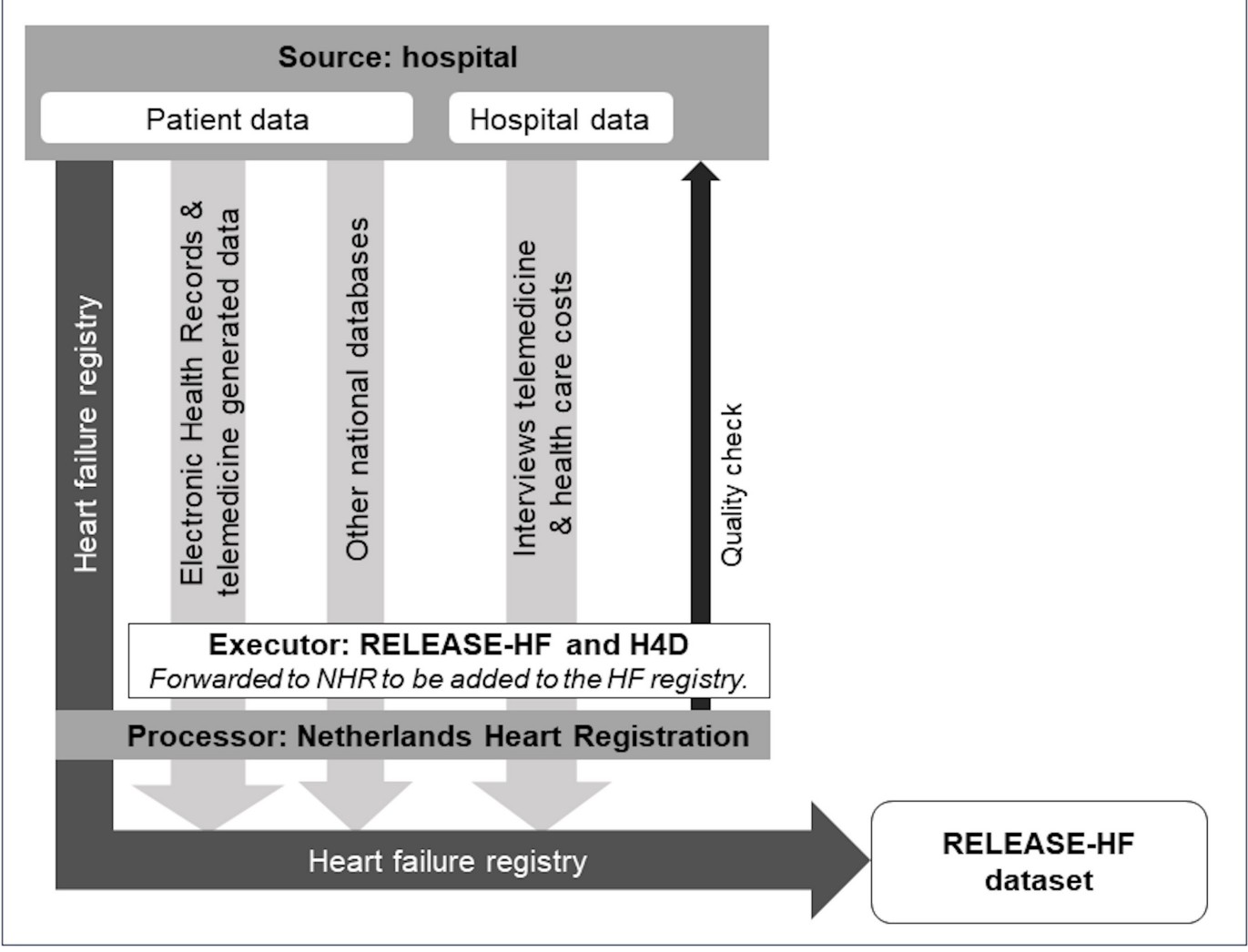

**Figure 2** Data sources and flow to set up the final RELEASE-HF data set. H4D, Heart4Data consortium; HF, heart failure; NHR, Netherlands Heart Registration; RELEASE-HF, REsponsible roLl-out of E-heAlth through Systematic Evaluation – Heart Failure.

(hrQoL), healthcare utilisation within 1 year of follow-up and costs of care (table 1).

### Exposure measurement

Exposure to telemedicine is measured in the HF registry at 6 months and 12 months of follow-up as non-use of telemedicine, telephone only, non-invasive telemedicine, implantable cardioverter defibrillator-based telemedicine or invasive telemedicine (table 1). Telemedicine output can be different from the closed-ended interviews that correspond to all outpatient clinic contacts in general. The components of the telemedicine intervention will be assessed at the hospital level at one time during the follow-up period. Table 2 provides an overview and operationalisation of the main components of the telemedicine intervention that have been described in existing literature or at the local (eg, hospital) level.[7 32–34] The components will be validated and possibly further refined based on the interviews. Figure 3 illustrates the flow of data collected at the hospital level and at the patient level.

### Additional variables

Additional information includes age at HF diagnosis, sex, social economic status (SES), body mass index (BMI), New York Heart Association (NYHA) classification, left ventricular ejection fraction (LVEF), aetiology of HF, systolic and diastolic blood pressure, and heart rhythm and rate. The comorbidities are chronic respiratory disorders, obstructive sleep apnoea syndrome (OSAS), stroke, extra cardiac arterial vascular pathology, diabetes mellitus (DM), hypertension, anaemia, chronic kidney disease (CKD), malignancy, heart rhythm disorders, depression and thyroid disease. An overview of these variables can be found in online supplemental material 1 and published in detail at the NHR website.[31] Table 3 provides an overview and operationalisation of variables derived from this HF registry.

### Data collection and quality control

Data of the HF registry will be remotely monitored for major outliers and missing variables using the NHR; these processes have been described elsewhere.[30 35] The NHR

van Eijk J, *et al*. *BMJ Open* 2024;**14**:e078021. doi:10.1136/bmjopen-2023-078021

**Table 1** Operationalisation of exposure variable and study outcomes

| Variable | Source | Time of collection | Operationalisation |
|---|---|---|---|
| **Exposure** | | | |
| Telemedicine | HF registry, telemedicine-generated data | T0, T1*, T2 | Whether a patient receives telemedicine. Five categories: (1) no telemonitoring†; (2) telemonitoring by telephone; (3) telemonitoring, non-invasive based on traditional parameters (eg, blood pressure, ECG); (4) telemonitoring using ICD based on HF parameters; and (5) telemonitoring, invasive using sensors in the bloodstream or heart. |
| **Primary outcome** | | | |
| Number of days spent alive without unplanned hospitalisations within 1 year of follow-up | HF registry | T1*, T2 | Number of days directly related to unplanned cardiac hospital admission due to HF. Number of admission days will be summed up over a period between follow-up moments and subtracted from 365 days. |
| **Secondary outcomes** | | | |
| Costs | Derived from HF registry, external data source, EHR, interviews | T0, T1*, T2 | Costs estimated from patient, disease and treatment characteristics. Information taken into account includes medication use, whether the patient underwent cardiac interventions (eg, pacemaker implantation, percutaneous coronary intervention), use of telemedicine, hospital admission days, visits to the outpatient clinic, visits to the emergency room, admission days at an intensive care unit and visits to the GP related to HF. |
| All-cause mortality | HF registry, external data source | T1*, T2 | Mortality status, determined after verification at the Personal Records Database (in Dutch: Basisregistratie Personen). Mortality is independent of HF (all-cause). |
| Functional status | HF registry | T0, T1*, T2 | NYHA classification: a functional classification of patients based on severity of symptoms and physical activity, with specific attention to fatigue, palpitation and dyspnoea. Scores are linked to one of four NYHA classes: class I: no limitation; class II: slight limitation; class III: marked limitation; and class IV: unable to carry on any physical activity without discomfort.[1] |
| Health status | HF registry | T0, T1*, T2 | SF-36 or SF-12 questionnaire (subset of SF-36)[54]: a validated patient-reported survey of patient health. Both questionnaires consist of eight sections with scores: vitality, physical functioning, bodily pain, general health perceptions, physical role functioning, emotional role functioning, social role functioning and mental health. Each score is transformed into a scale of 0–100 on the assumption that each question carries equal weight. The lower the score, the more disability. |
| Health-related quality of life | HF registry | T0, T1*, T2 | SF-36 or SF-12 questionnaire. QALY will be calculated based on the SF-6D, a model in which a single, preference-based score can be directly calculated for the SF-36 and SF-12.[55] Scores range from 0.0 (worst health state) to 1.0 (best health state). |
| Healthcare utilisation | External data source, EHR | T1, T2 | Healthcare utilisation based on (1) the number of outpatient visits, plus (2) the number of visits to the general practice related to HF. |

*The HF registry collects data standard at baseline (T0), after 6 months (T1) and after 12 months (T2). The RELEASE-HF study conforms to the timeframes of the HF registry. Therefore, data will be collected at 6 months, although the outcome measurements are after 12 months.
†The HF registry defines telemedicine as telemonitoring.
EHR, electronic health record; GP, general practitioner; HF, heart failure; ICD, implantable cardioverter defibrillator; NYHA, New York Heart Association; QALY, quality-adjusted life year; RELEASE-HF, REsponsible roLl-out of E-heAlth through Systematic Evaluation – Heart Failure; SF-12, 12-item short form health survey; SF-36, 36-item short form health survey; SF-6D, six-dimensional health state short form.

**Table 2** Overview and operationalisation of telemedicine variables collected in the qualitative study

| Component | Definition | Operationalisation (examples) |
|---|---|---|
| Supplier | (External) supplier of telemedicine for patients with HF. | Sanacoach, Luscii, Motiva/Philips, linked in personal environment in EHR. |
| Purpose of telemedicine | The intention/motives for which the telemedicine intervention is administered: patient level and/or hospital level. | Monitoring, prevent exacerbation, reduce workload, reduce costs, patient-centred care. |
| Considering telemedicine | First time a clinician considered telemedicine in HF management. | After diagnosis, (re)hospitalisation, titration phase. |
| Structured telephone support | Structured monitoring by telephone without using applications or devices specific to telemedicine and monitoring HF. | Present or not present. |
| Applications | Technologies or platforms on which the patient could receive telemedicine. | Smartphone, tablet, laptop, television, smartwatch. |
| Devices | Accessory a patient could use to perform telemedicine. | Blood pressure device (with or without Bluetooth), weight scale (with or without Bluetooth), smartwatch. |
| Involved healthcare workers | Involved healthcare workers and their role in considering and executing telemedicine. | HF nurse, nurse specialist, cardiologist. |
| Control centrum | The presence or absence of a control centrum to check the submitted measurements and questions. | Present or not present. |
| Use of telemedicine | When a patient could use telemedicine and have contact with clinicians. | 24/7, office hours, during weekend, at night. |
| Type of contact | The manner of contact between the clinician and the patient. | Direct or indirect (store-and-forward) contact with a clinician. |
| Measurements | Type of measurements: vital functions and HF-related complaints, used to detect deterioration of and/or to monitor HF. | Blood pressure, heart rate, weight, temperature, oxygen level, HF complaints such as swelling ankles, nocturia, shortness of breath, tiredness, loss of appetite, coughing/wheezing, dizziness. |
| Notifications | Messages from the telemedicine intervention. Notifications can be two-sided: from patient to clinician and vice versa. | Automatic or non-automatic generated messages; notifications present or not present. |
| Modifiable aspects | The option to set up *thresholds* of monitored vital function, and the possibility to tailor these thresholds per patient, severity, type of HF or other aspects of the telemedicine intervention (eg, set up a tele-education environment). | Available or not available. |
| Connection with EHR | Feature of telemedicine if the intervention is integrated with the EHR. This means that data entered by the patient through the telemedicine intervention are visible to the clinician in the EHR without using other applications. | Connected or not connected with EHR. |
| Education | The presence or absence of an educational environment and the manner in which this is shaped. | Present or not present. |
| Educational topics | The covered topics in the educational environment of telemedicine. | Nutrition, behaviour, exercise, medication. |
| Self-care | The presence or absence of self-care modules and the actions taken by the patient based on a digital advice or measurement. | Available or not available. |
| Protocol | Local (hospital) protocol which consists of definitions about how often a patient should use telemedicine/monitor the vital signs and HF complaints, depending on the HF complaints or phase (ie, titration, stable monitoring). | Low-intensity protocol defined as measurement of vital signs <2 times per week. High-intensity protocol defined as measurement of vital signs ≥2 times per week. |

EHR, electronic health record; HF, heart failure.

van Eijk J, *et al. BMJ Open* 2024;**14**:e078021. doi:10.1136/bmjopen-2023-078021

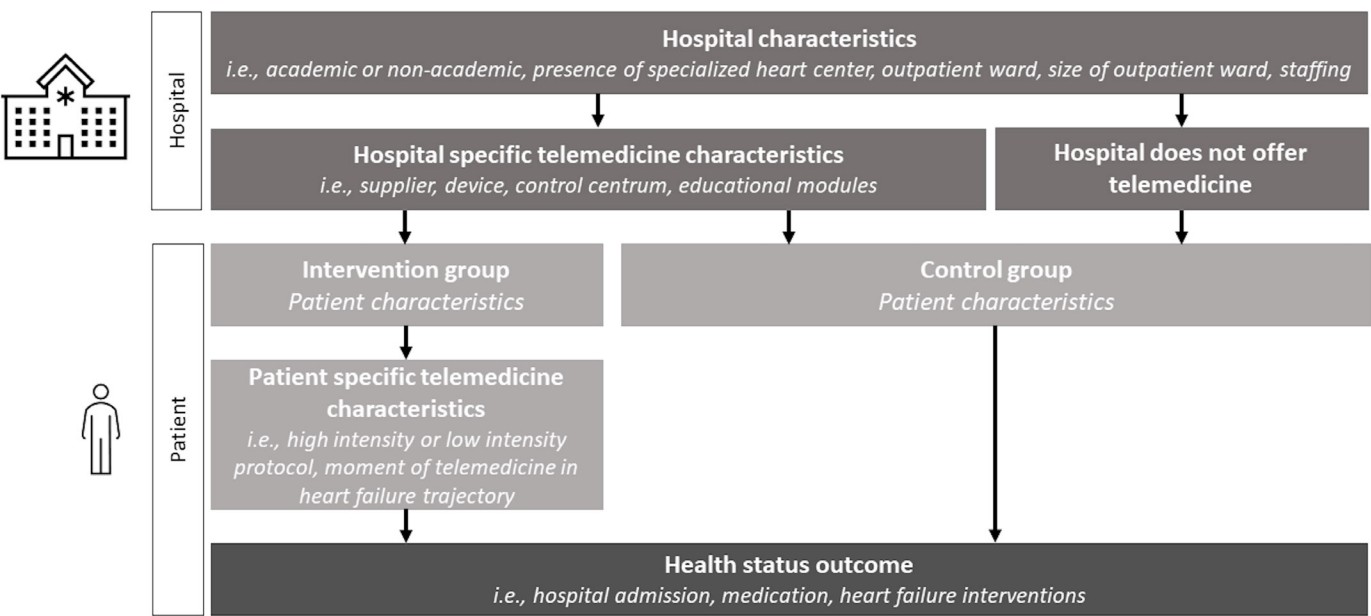

**Figure 3** Figure depicting how telemedicine characteristics are collected at the hospital level and at the individual patient level. It clarifies the source of the collected clinical content used for the subgroup analyses rather than the type of data source.

uses a unique individual patient identifier (NHRpersonID) to deduplicate registrations of the same patient (at the same or different hospitals) in the HF registry.[36] During the RELEASE-HF study, researchers will perform an additional check on completed data and provide assistance to hospitals to reduce missing values, as well as assistance to reduce registration burden.

Telemedicine interviews, using an interview guide with closed-ended questions, will be performed by a RELEASE-HF investigator and conducted with clinicians (eg, cardiologist, HF nurse, nurse specialist, physician assistant). Interviews will be held during the 36-month period of data collection. All hospitals with telemedicine in HF care will be interviewed once, and the data

will be checked with the hospitals once during the data collection period to be aware of changes in the components and status of telemedicine. For each hospital, the 6-monthly costs of telemedicine are determined via interviews and registration data (if at hand). The units will be linked to the unit cost guideline provided by the Dutch National Health Care Institute (Dutch: Zorginstituut Nederland).[37] These interviews will be performed by a health technology assessment (HTA) scientist and conducted with clinicians or finance department staff. All participants will provide informed consent.

Patient-level telemedicine data are collected by a clinician, data manager or researcher at the hospital where the patient is being treated because these individuals

| Table 3 | Operationalisation of variables calculated from the included variables in the HF registry |
|---|---|
| **Variable** | **Operationalisation** |
| Social economic status | Score for socioeconomic status (including degree of education, wealth and employment history (WOA)) by zip code area according to the 2019 data from the Social Cultural Planning Agency; divided into quartiles or quintiles, where 1 is a low SES-WOA score. |
| Body mass index | Derived from weight in kilogram and length in centimetres, calculated as weight/(length×length). |
| CKD | Based on the cut-off values of eGFR calculated from serum creatinine level (collected by the HF registry). The HF registry calculates eGFR as the following:<br>Male: $(175 \times (\text{creatinine level}/88.4)^{-1.154} \times (\text{age T}^{-0.203})$<br>Female: $(175 \times (\text{creatinine level}/88.4)^{-1.154} \times (\text{age T}^{-0.203}) \times 0.742$<br>CKD=eGFR $\geq 60 \,\text{mL/min}/1.73\,\text{m}^2$<br>No CKD=eGFR $< 60 \,\text{mL/min}/1.73\,\text{m}^2$ |
| Depression | Health-related QoL derived from the SF-12 or SF-36 questionnaire.<br>Scoring: 0–100; the lower the score, the more disability; the higher the score, the less disability; that is, a score of 0 is equivalent to maximum disability and a score of 100 is equivalent to no disability. |

CKD, chronic kidney disease; eGFR, estimated glomerular filtration rate; HF, heart failure; QoL, quality of life; SES, social economic status; SF-12, 12-item short form health survey; SF-36, 36-item short form health survey; WOA, wealth, educational level and employment history (in Dutch: welvaart, opleidingsniveau, arbeidsverleden).

have access to the identification log of the HF registry. Collected data will include the start date of telemedicine use and the frequency at which measurements are sent to the treating hospital, with a high rate typically referred to as a high-intensity (or acute) protocol and a low rate as a low-intensity (or stable) protocol.

The Heart4Data consortium will establish opportunities for data linkage between the HF registry and other national registry databases, in collaboration with public partners (figure 2).[28] At the time of writing, a collaboration between these databases and the HF registry is not yet established for all external national databases.

## Statistical analysis
### Clinical effectiveness

Descriptives are presented for all variables of the HF registry and telemedicine components. We will present the number of days without unplanned hospitalisation for HF during 365 days after baseline, the average duration of unplanned HF-related hospital stays, the number of deaths and the time until death.

### Preparation of the statistical model

As a first step, we describe how the overall effect of offering any type of telemedicine versus not offering telemedicine will be estimated. This analysis will subsequently be used for a comparison of telemedicine effects on days alive without unplanned HF-related hospitalisation during 1 year across subgroups.

The cohort is restricted to individuals with newly diagnosed HF, that is, at most 3 months prior to baseline. The exposure contrast is defined as offering any type of telemedicine after first presentation with HF at the hospital outpatient clinic versus patients not offered telemedicine. The telemedicine status for the entire follow-up is defined by 6 months after baseline measurement of telemedicine, used as an observational analogue of an 'intention to treat' analysis in which the stopping of telemedicine is disregarded.

Because telemedicine administration is dependent on patient characteristics, confounding by indication will be accounted for. Based on clinical consensus, the following patient-level confounding variables measured at baseline will be adjusted for: age, sex, SES, BMI, NYHA, LVEF, aetiology of HF, systolic blood pressure, diastolic blood pressure, heart rhythm, heart rate, chronic respiratory disorders, OSAS, stroke, extra cardiac arterial vascular pathology, DM, hypertension, anaemia, CKD, malignancy, heart rhythm disorders, depression and thyroid disease (see table 3 and online supplemental material 1).

The primary outcome model is a linear mixed model with a random intercept for the participating centre and is specified as follows. The outcome is measured in days, and because the final measurement may be less or more than 365 days after baseline, we rescale the number of days of unplanned hospitalisation to the number of days under follow-up to 365 days. The outcome is computed by subtracting the days of unplanned hospitalisation or death from 365. The earlier described confounding variables are added to the primary outcome model to adjust for confounding. Continuous covariates are modelled using splines. Multicollinearity between confounding variables and the telemedicine status is evaluated. The coefficient for telemedicine in the primary outcome model represents the average treatment effect of offering any telemedicine within 3 months after HF diagnosis compared with not offering telemedicine on days with unplanned HF-related hospitalisation during a year in the population of patients with newly diagnosed HF. Robust SEs are estimated to compute CIs.

For the secondary outcome functional status, a similar approach is taken to estimate the average treatment effect of offering any telemedicine within 3 months after HF diagnosis compared with not offering telemedicine on functional status after 1 year in the population of patients with newly diagnosed HF. For this analysis, the outcome model is a multinomial regression model, and baseline functional status is added as a covariate. The risk difference relative to the reference outcome category is assessed using the predicted outcomes of the primary outcome model offering of telemedicine versus not offering, to obtain the marginal risk difference.[38–40] Similarly, for the secondary outcomes health status and hrQoL, the outcome model is a probit regression model, with baseline measurements added as a covariate. Healthcare utilisation will be assessed using a Poisson regression model, and all-cause mortality will be assessed using binary logistic regression.

We plan to perform the following additional analyses. First, we plan to perform the analysis with the start of follow-up defined at the T1 (after 6 months) measurement and the outcome of unplanned hospitalisation days during 6 months. This is because telemedicine status can be misclassified when it is started shortly after baseline but is not registered in the data. Second, we plan to perform a per-protocol analysis of telemedicine use if detailed information on starting and stopping of telemedicine is available through linkage. Finally, we plan to perform all analyses in the entire cohort, rather than the cohort restricted on HF diagnosis, to a maximum of 3 months prior to T0 (baseline).

### Research question 1: effectiveness of telemedicine in patient subgroups

Research question 1 investigates the effectiveness of telemedicine across subgroups of patients with HF. Patient-related subgroups were identified by a systematic literature review of randomised controlled trials (RCTs) of telemedicine: age, severity of HF (NYHA class at baseline), sex, SES, presence of depression, atrial fibrillation and type of HF (HF with preserved ejection fraction (HFpEF), HF with midrange ejection fraction (HFmrEF), HF with reduced ejection fraction (HFrEF)) (table 3).[20–24 41–49] Stratified analyses of the primary outcome model are performed to estimate the above-defined average treatment effect in each of the

subgroups. Heterogeneity in telemedicine effect across age is assessed on a continuous scale, where an interaction between age and telemedicine status is added to the primary outcome model, and the expected number of days out of hospital across age ranges from 50 to 90 years is predicted from this model under telemedicine offered versus not offered.

In an additional analysis, heterogeneity across time since diagnosis is explored for the entire cohort, without restricting to patients with a maximum of 3 months since HF diagnosis at baseline (diagnosed ≤3 months before baseline compared with >3 months before baseline).

The secondary outcomes functional status, health status and hrQoL are assessed using similar approaches but using multinomial, probit and probit regression model, respectively. Healthcare utilisation will be assessed using a Poisson regression model, and all-cause mortality will be assessed using binary logistic regression.

### Research question 2: effectiveness of different telemedicine components

Research question 2 investigates the effectiveness of the different forms of telemedicine intervention. We assess this question by performing three separate analyses in which intervention aspects are contrasted in the subset of participants that received telemedicine. The presence or absence of a telemedicine component is determined at a hospital level. The components of telemedicine that are contrasted are presence versus absence of a service centre, presence versus absence of an educational module, and high-intensity versus low-intensity protocol (table 2).

The population of interest is patients with newly diagnosed HF (maximum 3 months prior to baseline) who received telemedicine. Offering of telemedicine is measured at 6 months after baseline.

In addition to the set of confounding variables at the patient level used in the analysis above, confounding variables at the hospital level are considered because indication for telemedicine is expected to differ across hospitals. Confounding variables include academic or non-academic hospital, presence of a specialised heart centre, presence of an outpatient ward, size of outpatient ward (number of patients/year), staffing of the HF outpatient clinic and full-time equivalent of outpatient-ward staff.

For each of the three contrasts, the primary outcome model is fitted to estimate the average treatment effect of a component relative to the corresponding reference component on days of unplanned hospitalisation during 1 year in the population of patients with newly diagnosed HF that initiated telemedicine.

The secondary outcomes functional status, health status and hrQoL are assessed using similar approaches but using multinomial, probit and probit regression model, respectively. Healthcare utilisation and all-cause mortality will be assessed using Poisson regression model and binary logistic regression, respectively.

### Cost-effectiveness

Costs will be estimated by computing the average costs (using cost guideline provided by the National Health Care Institute) per individual based on the information collected by the NHR and in the interviews with hospitals. Since 2023, Dutch hospitals can claim costs of telemedicine using the National Health Care Institute's Diagnosis Treatment Combination (DBC). This DBC can be used to estimate the average costs.[37] A 95% credible interval (CE) will be computed using the percentiles of a Monte Carlo bootstrap analysis with 5000 resamplings. This average cost represents a sum of the costs of telemedicine use, inpatient days, days at intensive care unit, HF-related hospital procedures and outpatient visits. In a secondary analysis, we will re-estimate the average costs and 95% CE using data enriched with VEKTIS data (if linking is available), meaning that the costs of visits to the general practitioner, pharmacy and care at home are also taken into account.

Linkage with CBS may be incomplete for several participants because of missing values or incomplete data sets (eg, twins, different available time windows between registries of collected data). If the number of non-linked participants is below 10%, we will perform multiple imputation; if it is above 10%, we will perform HTA analysis on the complete subset.

Utility values will be estimated using the average hrQoL score collected in the HF registry and, if needed, from the literature. The difference in average QoL score between the group that uses telemedicine versus the group that does not represents the difference in disutility between the groups under the assumption that the collected hrQoL is a good representation of the utility of the particular health state that individuals were in.

Quality-adjusted life years (QALYs) are estimated by multiplying the observed number of follow-up years by the corresponding hrQoL score. Subsequently, the incremental cost-effectiveness ratio (ICER) is computed by taking the ratio between the difference of the costs of the average patient in the telemedicine group and those not in the telemedicine group, and the difference in QALY of both groups, thereby providing the cost per additional QALY gained. The 95% CEs are obtained using the Monte Carlo percentile methods described above. Both deterministic and probabilistic sensitivity analyses will be performed to completely outline uncertainty on individual and combined parameters in the model.

In addition, willingness-to-pay curves will be drawn to highlight the impact of different thresholds on cost-effectiveness outcomes.

### Research question 3: cost-effectiveness of telemedicine in patient subgroups

For research question 3, costs, utility, QALYs, ICER and willingness-to-pay curves will be computed to estimate the difference in cost-effectiveness between users and non-users of telemedicine in general. Subsequently, the analyses will be repeated to estimate cost-effectiveness

in subgroups, similar to the groups defined for research question 2 in the clinical effectiveness analysis.

### Research question 4: cost-effectiveness of different telemedicine components

For research question 4, costs, utility, QALYs, ICER and willingness-to-pay curves will be computed to estimate the difference in cost-effectiveness between components of telemedicine, similar to the comparisons defined for research question 3 in the clinical effectiveness analysis.

### Sample size calculation

The primary focus in RELEASE-HF is on the clinical effectiveness of telemedicine in HF management. Hence, the sample size calculation is based on parameters relevant to this analysis. The sample size calculation was conducted in three steps: (1) computing the required sample size for the main effect of telemedicine on the primary outcome, (2) taking into account that subgroup effects are of primary interest and (3) anticipating the accrual rate.

For the first part, we assumed a mean increase in the number of days spent alive without unplanned hospitalisation of 6.4 (SD 21.1) days based on the findings of the TIM-HF2 study.[44] This corresponds to a required sample size of 432 patients in total, based on a type I error probability of 5%, a type II error probability of 20% and accounting for 20% dropout.

Such a sample size would allow for an overall estimate of the difference between telemedicine and no telemedicine in days spent alive without unplanned hospitalisation in the overall patient population; however, our interest is in heterogeneous intervention effects. Hence, for the second part, the sample size was inflated to estimate subgroup effects.[50 51] To detect interaction effects that are 50%–60% of the size of the overall effect (ie, increase of more than 3.2–3.8 days spent outside the hospital due to the interaction given the main effect) with a power of 80%, the required sample size is inflated by a factor of 12.[50] This results in a required sample size of 5184 individuals.

Finally, from the HF registry pilot study (CHECK-HF), we know that, in view of the estimated average number of patients per hospital, the average proportion of included outpatient clinic patients with HF was above 80%.[52] Taking this accrual rate into account, we would require the inclusion of 6480 patients with HF.

### Patient and public involvement

Patient and public organisations were involved during grant application. Public organisations were involved in recruiting hospitals, and in legal support and advice. Healthcare professionals' involvement in the study includes participating in an interview about telemedicine and motives about telemedicine choice and use. Patients with HF and patient organisations will be involved in formulating and prioritising relevant research questions which are in line with the need of the patient and the aim of the RELEASE-HF study. The results will be disseminated with involvement of patient and public organisations.

## ETHICS AND DISSEMINATION
### Management and storage of data

RELEASE-HF is an observational, retrospective multicentre cohort study of prospectively collected data registered within the NHR. A waiver for informed consent for analysis of data from the NHR data registry was obtained. Data collection and registration is performed by the participating centres in a secured online environment. Detailed information on the process of data acquisition, completeness, data quality and analysis of the NHR has been published previously.[30] To obtain reliable data, the NHR has an advanced, certified data quality control system in place to ensure completeness and quality of data.[35]

Participants of the interview study will sign informed consent. Data will be stored in the secured environment at the University Medical Center (UMC) Utrecht, the Netherlands. Data will be pseudonymised at the participant and setting level. A secured identification log will be used, only accessible to the main RELEASE-HF researcher. Because the interview data will be linked to the HF registry, pseudonymised data will also be stored at the Medical Informatics Department of University Medical Centers Amsterdam (the Netherlands).

Storage of linked data sets (HF registry with other national databases) will be part of the infrastructure established by the Heart4Data consortium.[28] The RELEASE-HF study will follow these principles.

The Medical Ethics Committee of UMC Utrecht (the Netherlands) reviewed the study protocol and confirmed that the study does not fall under the scope of the Medical Research Involving Human Subjects Act. RELEASE-HF complies with the rules of the General Data Protection Regulation.

### Dissemination

The results will be published in peer-reviewed journals and presented at (inter)national conferences as deemed relevant for HF and telemedicine. The HF registry data underlying this article were provided by the NHR with the permission of the participating hospitals. Data are available on reasonable request to the corresponding author and with permission of the NHR. The hospital-specific telemedicine characteristics, which are added to the HF registry data, will also only be available on reasonable request to the corresponding author and with permission of the participating hospitals.

## DISCUSSION

The RELEASE-HF study is a large-scale, observational study used to better understand heterogeneity in clinical effectiveness between patients using telemedicine. To our knowledge, this is the first study using routine clinical care data.

The current design has been chosen because telemedicine is already implemented in various ways in healthcare settings in the Netherlands. Therefore, conducting an RCT does not fit the current care for patients with HF. Additionally, using a national registry instead of an RCT allows us to observe all patients with HF, reducing selection bias that would otherwise be introduced by an RCT. A Dutch registry study previously showed that routinely collected data lead to a representative sample.[52] However, we have to be aware of confounding bias introduced by physicians or nurses who decide which patients may use telemedicine. We cannot completely avoid this bias, although we perform extensive confounding correction and conduct interviews with these healthcare professionals, asking about their local guidelines so we understand the selection of patients in that hospital. Consequently, it has been argued that results from observational studies describe a patient outcome when using the intervention rather than assessing the response to the intervention.[53] Another limitation of registry-based studies is that the data to be collected are predetermined, which can lead to relevant missing variables. To overcome the missing data, such as telemedicine data at the patient level or outcome measures, we aim to link other data sources to enrich the RELEASE-HF data set.

## Author affiliations

[1] General Practice and Nursing Science, Julius Center for Health Sciences and Primary Care, University Medical Center Utrecht, Utrecht, The Netherlands
[2] Epidemiology and Health Economics, Julius Center for Health Sciences and Primary Care, University Medical Center Utrecht, Utrecht, The Netherlands
[3] Netherlands Heart Registration, Utrecht, The Netherlands
[4] Department of Cardiology, Amphia Hospital, Breda, The Netherlands
[5] Dutch Network for Cardiovascular Research, WCN, Utrecht, The Netherlands
[6] Department of Cardiology, Erasmus Medical Center, Rotterdam, The Netherlands
[7] Department of Cardiology, Amsterdam University Medical Centers, Amsterdam, The Netherlands
[8] Health Data Research UK and Institute of Health Informatics, University College London, London, UK
[9] The Healthcare Innovation Center, Julius Center for Health Sciences and Primary Care, University Medical Center Utrecht, Utrecht, The Netherlands

**Acknowledgements** We would like to acknowledge all the study participants, cardiologists and nurses from all participating hospitals for setting up the RELEASE-HF study and the data collection in their hospitals.

**Collaborators** RELEASE-HF Investigators: C Jan Willem Borleffs, Dirk H van Dalen, Ayten Erol-Yilmaz, M Louis Handoko, Gerardus PJ van Hout, Jaco Houtgraaf, Wouter W Jansen Klomp, Stefan Koudstaal, Gerard CM Linssen, Manon G van der Meer, Marco C Post, Sandra Sanders-van Wijk, Eric Wierda, Stijn CW Wouters.

**Contributors** JvE contributed to protocol conception and design and to the development of statistical analyses, obtained ethics approval, and contributed to writing the protocol manuscript and critical revision of the protocol manuscript. KL contributed to protocol conception and design and to the development of statistical analyses, provided statistical and methodological support, and contributed to writing the statistical analyses paragraph and critical revision of the protocol manuscript. JBR contributed to protocol conception and design, obtaining funding, and development of statistical analyses, provided statistical and methodological support, and critically revised the manuscript. ES contributed to the development of statistical analyses, provided statistical and methodological support, and critically revised the manuscript. GWJF contributed to protocol conception and design, obtaining funding, and writing the economic evaluation paragraph, and critically revised the manuscript. LD contributed to protocol conception for data collection combined with the use of the HF registry data, and critically revised the manuscript. JS and JB are also local investigators and assisted with the data collection. TJ, JS, JB, FHR and RAdB contributed to protocol conception and design and obtaining funding, and critically revised the manuscript. FWA and JCAT contributed to protocol

conception and design, secured funding for the project and critically revised the manuscript. All authors approved the final version of the manuscript. The RELEASE-HF investigators consist of all local investigators who are responsible for ethical board approval and data collection. They have all read, refined and approved the final version of the manuscript.

**Funding** This study is supported by the Netherlands Organization for Health Research and Development (ZonMw grant 852002141). Additional support has been received from the Dutch CardioVascular Alliance (DCVA). Hospitals receive €40 per included patient, paid by the RELEASE-HF study, with a complete 1-year follow-up (ie, data collected at three timepoints: at baseline, 6 months and 12 months from time since diagnosis). This funding is only related to hospitals participating in the RELEASE-HF study and will stop when the study is complete. The HF registry does not pay hospitals to participate in the registry.

**Competing interests** JB received an independent research grant from Abbott for ISS paid to institution, and has had speaker engagements or advisory boards for AstraZeneca, Abbott, Bayer, Boehringer, Daiichi Sankyo, Novartis and Vifor in the past 5 years. RAdB has received research grants and/or fees from AstraZeneca, Abbott, Boehringer Ingelheim, Cardior Pharmaceuticals, Ionis Pharmaceuticals, Novo Nordisk and Roche, and has had speaker engagements with Abbott, AstraZeneca, Bayer, Bristol Myers Squibb, Novartis and Roche. CJWB has received speaker fees from AstraZeneca, Boehringer Ingelheim and Novartis. MLH is supported by the Dutch Heart Foundation (Dr E Dekker Senior Clinical Scientist Grant 2020T058) and CVON (2020B008 RECONNEXT). He received an investigator-initiated research grant from Vifor Pharma, an educational grant from Boehringer Ingelheim and Novartis, and speaker/consultancy fees from Abbott, AstraZeneca, Bayer, Boehringer Ingelheim, MSD, Novartis, Sankyo, Daiichi, Quinn and Vifor Pharma, all of which were not related to this study.

**Patient and public involvement** Patients and/or the public were involved in the design, or conduct, or reporting, or dissemination plans of this research. Refer to the Methods section for further details.

**Patient consent for publication** Not required.

**Provenance and peer review** Not commissioned; externally peer reviewed.

**ORCID iDs**
Jorna van Eijk http://orcid.org/0000-0003-1251-6736
Frans H Rutten http://orcid.org/0000-0002-5052-7332

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
