## [Reviewer comments · BMJ Open]

ARTICLE DETAILS

TITLE (PROVISIONAL)	RELEASE-HF study protocol: An observational, registry-based study on the effectiveness of telemedicine in heart failure in the Netherlands
AUTHORS	van Eijk, Jorna; Luijken, Kim; Jaarsma, Tiny; Reitsma, Johannes; Schuit, Ewoud; Frederix, Geert; Derks, Lineke; Schaap, Jeroen; Rutten, Frans; Brugts, Jasper; de Boer, Rudolf; Asselbergs, Folkert; Trappenburg, Jaap; investigators, RELEASE-HF

VERSION 1 – REVIEW

REVIEWER	iyngkaran, pupalan Royal Darwin Hospital, Cardiology
REVIEW RETURNED	02-Sep-2023

GENERAL COMMENTS	Thank you for the opportunity to review this submission. A multicenter nationwide study (in any area) in Telemedicine in HF is very important addition to the literature. In this case from Netherlands, which has good health systems in place. So I think this paper and its findings has importance. My comments' are as follow: 1. However, despite the positive comments this paper cannot be accepted in the current form. From my reading of the title and abstract alone the english language and sentencing is poor, in parts. e.g the way the words are used to describe the key ingredients "Study Protocol" "Registry" "National" (does it need multicentre?). Heart failure is it acute, chronic, reduce, preserved. Do we need the word patients? Please find a sharp punchy title2. Abstract: "Telemedicine gradually gains acceptance in heart failure (HF) management and meta-analyses show positive effects of telemedicine on hospitalization, mortality, and costs." is a very clumsy sentence. Please rewrite the abstract language and sentencing.3. Keywords and limitations - starting to show then focus and strength of the study and data.4. Introduction - very good. References are current. Data content is good and accurate. Again some major language e.g- "economic and social .." vs socioeconomic- "Optimal medical management, but also self-care". I prefer Optimised Guideline driven medical therapy (GDMT) including good self-management (self-care can have different meaning).- "Recently, interest ..." vs The burden of HF and health systems has seen health models utilise telemedicine to support ambulatory .HF management strategies ... Telemedicine has been shown5. Methods- Aims - please check with editorial team, but good to put study aims early and as several points. Can be at end of introduction or beginning of methods.
---

	- Design - "...observational character.." is not a scientific term. Please use a definitive term to describe. Figures and table - suggest very strong processes!! The rest of the manuscript is very strong. Summary: I am going to recommend a major review based on language mainly. I think it is hard to read the rest in earnest, thus it may be another 1-2 reviews before acceptance. (my view) But a good study we should support.
--	--

REVIEWER	Cleland , J Glasgow University , School of Cardiovascular & Metabolic Health
REVIEW RETURNED	04-Sep-2023

GENERAL COMMENTS	This is a design paper for an observational study of telemonitoring for heart failure in the Netherlands. Because it is observational, it will be impossible to reliably distinguish between selection bias (by the doctor, patient or local provision) and the effects of intervention. Only limited conclusions can be drawn from this exercise. The limitations of observational analyses need to be emphasized more clearly. A separate section devoted to limitations is needed. Figure 1 is important for orientation. It should be made clearer that all patients should be included at first encounter whether or not they are offered and accept telemonitoring. Presumably some patients will be registered more than once (either in the same or different hospitals). How is this dealt with? Presumably patients who are unable to cope with telemonitoring (or who have conditions more serious than heart failure) will be less likely to be offered telemonitoring. This should be added to limitations. The authors say "Most data on telemedicine characteristics will be collected on a hospital level rather than on the individual patient level" but surely the nature of the telemonitoring offered to individual patients is important. Please make this clearer when reporting how the study is conducted. Title This should contain the word 'observational' to orientate readers and searches. Introduction The first paragraph is unnecessary, superficial and best deleted. The second paragraph is rather repetitive and should be re-written. There are some issues with English expression that could be sorted by the editorial office. Methods Enrolment is unclear. I think they mean to enrol as many patients as possible into a national registry, whether or not they are enrolled in a telemedicine programme. It is not clear what the diagnostic criteria for heart failure are (there are substantial differences in opinion and in case ascertainment). The authors should be able to provide some idea of the age and sex of patients
--

	enrolled so far and provide some idea of the proportion of potential patients enrolled. This should be >1% of the surrounding community. It is not yet clear whether the authors will be able to link their patients to other national data-sources. Without this, the study is of much less value because it is unlikely that re-hospitalisation data will be accurately entered by site staff. Funding A statement on how much sites were reimbursed per patient would be valuable as a measure of the resource available for data-recording (which will be a surrogate for quality). Table 2 Region and distance from centre are also important variables that will have been collected and should be used for analyses. Clinical effectiveness - the limitations of trying to assess this in observational data should be highlighted. Associations between telemedicine use and outcome may not be causal. In the multi-variable analysis it will be important to look for and manage problems arising from collinearity. Surprised not to see ventricular paced amongst the choice of heart rhythms. Units for NT-proBNP are usually ng/L or pg/mL. The criteria for iron deficiency are likely to change soon (see Ponikowski et al Eur Heart J August 2023). There are many classes of diuretic of which loop diuretic is the most important for heart failure. It is critical that the authors gather and report data by diuretic class.
--	---

VERSION 1 – AUTHOR RESPONSE

Reviewer: 1

Dr. pupalan iyngkaran, Royal Darwin Hospital, Flinders University

Comments to the Author:

Thank you for the opportunity to review this submission. A multicenter nationwide study (in any area) in Telemedicine in HF is very important addition to the literature. In this case from Netherlands, which has good health systems in place. So I think this paper and its findings has importance.

Response:

We thank the reviewer for acknowledging the importance of our study.

My comments' are as follow:

1. **However, despite the positive comments this paper cannot be accepted in the current form. From my reading of the title and abstract alone the English language and sentencing is poor, in parts. e.g the way the words are used to describe the key ingredients "Study Protocol" "Registry" "National" (does it need multicentre?). Heart failure is it acute, chronic, reduce, preserved. Do we need the word patients? Please find a sharp punchy title**

Response: We thank the reviewer for their advice to improve the language of our manuscript. We have edited and improved the English throughout. We agree that a title including the key ingredients is more sharp and punchy, and also meets the expectations of readers. We have reworded the title, also incorporating suggestions from the editor and other reviewer.

ADJUSTMENT:

'RELEASE-HF study protocol: An observational registry-based study on the effectiveness of telemedicine in heart failure in the Netherlands.'

2. **Abstract: "Telemedicine gradually gains acceptance in heart failure (HF) management and meta-analyses show positive effects of telemedicine on hospitalization, mortality, and costs." is a very clumsy sentence. Please rewrite the abstract language and sentencing.**

Response: We agree that the first sentence of the abstract can be improved. Therefore, we separated this sentence into two sentences. Additionally, we edited and improved some sentencing throughout the abstract.

ADJUSTMENTS:

Page 3:

Meta-analyses show positive effects of telemedicine in heart failure (HF) management on hospitalization, mortality, and costs. However, these effects are heterogeneous due to variation in the included HF population, telemedicine components, and the quality of the comparator usual care. Still, telemedicine is gaining acceptance in HF management. The current nationwide study aims to identify (i) in which subgroup(s) of HF patients telemedicine is (cost-)effective, and (ii) which components of telemedicine are most (cost-)effective.

The 'REsponsible roLI-out of E-heAlth through Systematic Evaluation – Heart Failure' study (RELEASE-HF) is a multi-center, observational, registry-based cohort study planning to enroll 6,480 HF patients using data from the HF registry facilitated by the Netherlands Heart Registration.

Collected data include patient characteristics, treatment information, and clinical outcomes and are measured at HF diagnosis, 6 and 12 months afterwards.

Components of telemedicine are described at hospital level based on close-ended interviews with clinicians and at patient level based on additional data extracted from Electronic Health Records and telemedicine generated data.

To overcome missing data additional national databases will be linked to the HF registry if feasible.

Heterogeneity of effects of offering telemedicine compared to not offering on days alive without unplanned hospitalizations in 1 year are assessed across predefined patient characteristics using exploratory stratified analyses.

Effects of telemedicine components are assessed by fitting separate models for component contrasts.

3. Keywords and limitations - starting to show then focus and strength of the study and data.

Response: We suppose the reviewer means that the strength and limitation section should be in a different order. However, we are concerned that a different order might confuse readers and decided to keep the order of strengths and limitations as it is.

4. Introduction - very good. References are current. Data content is good and accurate. Again some major language e.g.

- a. "economic and social .." vs socioeconomic
- b. "Optimal medical management, but also self-care". I prefer Optimised Guideline driven medical therapy (GDMT) including good self-management (self-care can have different meaning).
- c. "Recently, interest ..." vs The burden of HF and health systems has seen health models utilise telemedicine to support ambulatory .HF management strategies ... Telemedicine has been shown

Response: We thank the reviewer for their positive appraisal of the introduction. We appreciate the suggestion to improve the language. We edited and improved the English throughout the introduction.

Regarding the use of self-management instead of self-care, we preferred to use the wording from the ESC guidelines, that is 'self-care' and is used in heart failure (HF) management literature (Jaarsma et al., 2021, EJHF). The concept self-care represents the aspect central to our research, where self-care is defined as a process of maintaining health through health promoting and preventive practices. It consists of three components: maintenance, monitoring and management. In our opinion, these components are also components of telemedicine.

ADJUSTMENTS:

Page 5: Heart failure (HF) poses a major socio-economic and patient burden. Healthcare systems are seeking innovative health models to support care. Optimized guideline-directed medical therapy (GDMT), self-care (i.e., healthy diet, medication adherence, exercise), and adequate monitoring of vital signs and symptoms may all help to reduce HF-related morbidity.[1-3]. Health models therefore use telemedicine as a tool to support patients in optimizing HF management, self-care support and symptom monitoring to improve care and prevent (re-)hospitalization.[2-4] "Telemedicine" is an heterogeneous intervention that includes a wide range of digital technologies (smartphones, mobile wireless devices/sensors, video connections, implantable devices, etc.) exchanging digital health information between patient and clinician to support and optimize the care process remotely.[5, 6]

Many meta-analyses have evaluated the (cost-)effectiveness of telemedicine for HF patients.[7-19] Overall, these meta-analyses point towards a positive effect of telemedicine on hospital admission, length of hospital stay, mortality, and costs.[2, 7-19] However, effects are heterogeneous across subgroups [20-24], and vary with telemedicine components, such as

type of monitored functions, number of alerts and risk of alert fatigue, contact with health care professional, type of device, and different quality of usual care.[7, 14, 15, 19, 25] This can lead to problems in comparability of studies. Understanding of the cost-effectiveness of telemedicine is fragmented because studies do not evaluate the same costs, with the focus shifting between the purchase costs of telemedicine, personnel costs, and the variation in the costs of telemedicine components.[4, 12, 16]

As a result, HF guidelines lack specific recommendations on how, when and in whom telemedicine should be provided.

5. Methods

- a. **Aims - please check with editorial team, but good to put study aims early and as several points. Can be at end of introduction or beginning of methods.**
- b. **Design - "...observational character.." is not a scientific term. Please use a definitive term to describe.**
- c. **Figures and table - suggest very strong processes!!**
- d. **The rest of the manuscript is very strong.**

Response: We appreciate the compliments of the reviewer. The study aims are stated at the end of the introduction and reiterated at their respective statistical analysis. The aims in the introduction supports the reader in getting an idea of the overall study purpose and the aims in the statistical analysis ensure a link to the specific methods.

ADJUSTMENT:

Page 6: RELEASE-HF is a nationwide, observational, registry-based cohort study across multiple hospitals in the Netherlands.[27] The study collects routine data longitudinally.

Reviewer: 2

Dr. J Cleland , Glasgow University , University of Glasgow

Comments to the Author:

1. **This is a design paper for an observational study of telemonitoring for heart failure in the Netherlands. Because it is observational, it will be impossible to reliably distinguish between selection bias (by the doctor, patient or local provision) and the effects of intervention. Only limited conclusions can be drawn from this exercise. The limitations of observational analyses need to be emphasized more clearly. A separate section devoted to limitations is needed.**

Response: Thank you for reviewing our manuscript. We are aware of the limitations but also the strengths of using an observational study design in answering the formulated research questions. Because of the current use and ongoing roll-out of telemedicine in the Netherlands, it is not feasible to initiate a new nationwide randomized controlled trial. In our view, this new reality requires a different methodological approach using real-world data. In the design of our study, we considered different mechanisms to limit selection bias and apply

corrections for confounding, as described in the data collection and methodology section. Although the presence of residual confounding can never be excluded, this study is a best effort to unravel the effectiveness of telemedicine as currently implemented in the Netherlands in a broad population of HF patients.

To the main text, we added a discussion section to make the limitations and strengths clearer instead of only mentioning it in the strength and limitation section at the beginning of the manuscript.

ADJUSTMENT:

Page 16: DISCUSSION

The RELEASE-HF study is unique because of its observational study design, used to better understand heterogeneity in clinical effectiveness. To our knowledge, this is the first study using real-world data. The current design is chosen, because telemedicine is already implemented in various ways in health care settings in the Netherlands. Therefore, conducting a randomized controlled trial (RCT) does not fit the current care for HF patients. Additionally, using a national registry instead of a RCT allows us to observe all HF patients reducing selection bias that would otherwise be introduced by a RCT. A Dutch registry study previously showed that routinely collected data lead to a representative sample.[54] However, we have to be aware of confounding bias introduced by physicians or nurses who decide which patients may use telemedicine. We cannot completely avoid this bias, although we perform extensive confounding correction and conduct interviews with these health care professionals and asking about their local guidelines to understand the selection of patients in that hospital. Another limitation of registry-based studies is that the data to be collected are predetermined, which can lead to relevant missing variables. To overcome the missing data, such as telemedicine data on patient level or outcome measures, we aim to link other data sources to enrich the RELEASE-HF dataset.

- 2. Figure 1 is important for orientation. It should be made clearer that all patients should be included at first encounter whether or not they are offered and accept telemonitoring.**

Response: We thank the reviewer for pointing out this unclarity in Figure 1. The figure was clarified and now mentions that all patients, whether they received telemedicine, are included in the HF registry.

ADJUSTMENT:

3. Presumably some patients will be registered more than once (either in the same or different hospitals). How is this dealt with?

Response: The reviewer raises an important point about how a national registry operates. Thank you for your comment. According to the policy of the Netherlands Heart Registration (NHR), each individual patient included in the registry has a personal NHR identifier. This identifier consists of a combination of variables: age, sex, and the first four letters of the birth name. The identifier is used to deduplicate patients who are included in the registry more than once, at the same or different hospital(s). In addition, it is registered when a patient is referred from a hospital to another. The RELEASE-HF includes only patients after initial diagnosis.

ADJUSTMENT:

Page 11: The NHR uses a unique individual patient identifier (NHRpersonID) to deduplicate registrations of the same patient (at the same or different hospitals) in the HF registry.[38]

4. Presumably patients who are unable to cope with telemonitoring (or who have conditions more serious than heart failure) will be less likely to be offered telemonitoring. This should be added to limitations.

Response: We thank the reviewer for pointing out this limitation. In this study we have to deal with this form of selection bias. Through interviews (if needed and possible also checks of local, i.e., hospital, guidelines), we try to determine for each hospital which patients are eligible for telemedicine and which patients are not. This information will be used in the statistical analyses to correct for confounding by indication. We added a discussion paragraph to the main text covering these selection bias.

ADJUSTMENT:

The RELEASE-HF study is unique because of its observational study design, used to better understand heterogeneity in clinical effectiveness. To our knowledge, this is the first study using real-world data. The current design is chosen, because telemedicine is already implemented in various ways in health care settings in the Netherlands. Therefore, conducting a randomized controlled trial (RCT) does not fit the current care for HF patients. Additionally, using a national registry instead of a RCT allows us to observe all HF patients reducing selection bias that would otherwise be introduced by a RCT. A Dutch registry study previously showed that routinely collected data lead to a representative sample.[54] However, we have to be aware of confounding bias introduced by physicians or nurses who decide which patients may use telemedicine. We cannot completely avoid this bias, although we perform extensive confounding correction and conduct interviews with these health care professionals and asking about their local guidelines to understand the selection of patients in that hospital. Another limitation of registry-based studies is that the data to be collected are predetermined, which can lead to relevant missing variables. To overcome the missing data, such as telemedicine data on patient level or outcome measures, we aim to link other data sources to enrich the RELEASE-HF dataset.

- 5. The authors say "Most data on telemedicine characteristics will be collected on a hospital level rather than on the individual patient level" but surely the nature of the telemonitoring offered to individual patients is important. Please make this clearer when reporting how the study is conducted.**

Response: We agree with the reviewer and meant this as a limitation of our study. We clarified this by stating this explicitly.

ADJUSTMENTS:

Page 4: A strength of this study is using a nationwide registry-based cohort linked to national registries and databases to capture real world data on telemedicine use, heart failure patient characteristics, treatments, and clinically relevant outcomes with a 1-year follow-up.

As another strength, data of the Heart failure registry in the Netherlands and a separate data collection on telemedicine features by additional questionnaires allow consideration of heterogeneous treatment effects across the heart failure population and telemedicine components.

As a limitation, most data on telemedicine characteristics will be collected on a hospital level rather than on the individual patient level.

As another limitation, cost-effectivity analyses are specific for the Dutch healthcare system, which may hamper generalizability of findings to other settings or countries.

A final limitation is that selection bias can occur when healthcare providers do not include all patients in the outpatient clinic in the Heart failure registry, because the Heart failure registry is not incorporated in care pathways in Electronic Health Record systems."

- 6. Title
This should contain the word 'observational' to orientate readers and searches.**

Response: We added the word *observational* to the title.

ADJUSTMENT:

'RELEASE-HF study protocol: An observational registry-based study on the effectiveness of telemedicine in heart failure in the Netherlands.'

7. Introduction

The first paragraph is unnecessary, superficial and best deleted.

The second paragraph is rather repetitive and should be re-written.

There are some issues with English expression that could be sorted by the editorial office.

Response: We reread the introduction and agree that the first paragraph should be deleted. The second paragraph is rewritten. Additionally, we edited and improved some sentencing and English expressions throughout the introduction.

ADJUSTMENTS:

Page 5: Heart failure (HF) poses a major socio-economic and patient burden. Healthcare systems are seeking innovative health models to support care. Optimized guideline-directed medical therapy (GDMT), self-care (i.e., healthy diet, medication adherence, exercise), and adequate monitoring of vital signs and symptoms may all help to reduce HF-related morbidity.[1-3]. Health models therefore use telemedicine as a tool to support patients in optimizing HF management, self-care support and symptom monitoring to improve care and prevent (re-)hospitalization.[2-4] "Telemedicine" is an heterogeneous intervention that includes a wide range of digital technologies (smartphones, mobile wireless devices/sensors, video connections, implantable devices, etc.) exchanging digital health information between patient and clinician to support and optimize the care process remotely.[5, 6]

8. Methods

Enrolment is unclear. I think they mean to enrol as many patients as possible into a national registry, whether or not they are enrolled in a telemedicine programme. It is not clear what the diagnostic criteria for heart failure are (there are substantial differences in opinion and in case ascertainment). The authors should be able to provide some idea of the age and sex of patients enrolled so far and provide some idea of the proportion of potential patients enrolled. This should be >1% of the surrounding community.

Response: Patients are included in the RELEASE-HF study through the national HF registry. This registry is still new, so the RELEASE-HF study is helping to commit hospitals to work with the national registry. When the RELEASE-HF study ends, the HF registry will continue to exist. Because we use the HF registry as a tool to collect data, RELEASE-HF is enrolling as many patients as possible, whether or not enrolled in a telemedicine programme (we added this to Figure 1 in the manuscript). We also need data of *control patients* (i.e., patients not using telemedicine) to make statements about the clinical effectiveness of telemedicine in patients enrolled to a telemedicine programme.

To include patients in the HF registry, they need to meet the diagnostic criteria for HF and its phenotype according to the ESC 2021 guidelines. In the manuscript we described this in the

paragraph study population. In the main text we adjusted the sentence to make clear that the diagnostic criteria for HF is based on the ESC HFA guideline.

From a Dutch pilot study that used the national HF registration, we can conclude that the registration is representative for the Dutch HF patients. In this study (n=10.910 patients) the average age was 72.8 years, and around 60% was male (Brugts et al. 2018, NHJ, CHECK-HF study).

ADJUSTMENT:

Page 7: The HF registry comprises patients admitted to a Dutch hospital or outpatient clinic for HF irrespective of telemedicine use (figure 1). All patients who meet the diagnostic criteria for HF and its phenotype according to the ESC 2021 guidelines on HF are included.[1]

- 9. It is not yet clear whether the authors will be able to link their patients to other national data-sources. Without this, the study is of much less value because it is unlikely that re-hospitalisation data will be accurately entered by site staff.**

Response: Thank you for expressing your concern and what it might mean if we do not receive approval to link the data. We share your concern. However, the Heart4Data consortium is making progress to enable data linkages. Currently, linkage to data of Statistics Netherlands (CBS) is possible through approval from the HF committee of the NHR. Additionally, the Heart4Data consortium is in the final stages of negotiating with PHARMO and Dutch Hospital Data (DHD) so that data linkage will soon be legally possible with these data sources. Technically, data linking is already possible. Through data linkage with CBS and/or DHD, it is possible to receive more accurate data on re-hospitalisation.

10. Funding

A statement on how much sites were reimbursed per patient would be valuable as a measure of the resource available for data-recording (which will be a surrogate for quality).

Response: We added a statement about the reimbursement per patient in the paragraph about funding provided by the RELEASE-HF study.

ADJUSTMENT:

Page 22: Hospitals receive 40 euros per included patient, paid by the RELEASE-HF study, with a complete one-year follow-up (i.e., data collected at three timepoints: at baseline, 6 months, and 12 months from time since diagnosis). This funding is only related to hospitals participated in the RELEASE-HF study, and will stop after finishing this study. The HF registry do not pay hospitals to participate in the registry.

11. Table 2

Region and distance from centre are also important variables that will have been collected and should be used for analyses.

Response: The addition to collect data on region and distance of the patient from the hospital is of value in a study about telemedicine. Unfortunately, in RELEASE-HF we cannot collect

these data because we are following the national HF registry, and in the Netherlands it's not legal to receive this personal data without informed consent of the patient (i.e., there is no informed consent in this study since RELEASE-HF reuse data from a national HF registry). This is against the *General Data Protection Regulation (GDPR)*. Statistics Netherlands investigated the average distance to a hospital in the Netherlands. They conclude that all inhabitants reach a hospital within 30 minutes, and the average distance is less than 5 kilometres. Therefore, its questionable if these variables, region and distance, is of the same importance compared to other and bigger countries.

12. Clinical effectiveness - the limitations of trying to assess this in observational data should be highlighted. Associations between telemedicine use and outcome may not be causal. In the multi-variable analysis it will be important to look for and manage problems arising from collinearity.

Response: The reviewer is correct in pointing out the difficulty of using an observational study design to answer clinical effectiveness and refer to point 1 to motivate why we are using observational data to address causal questions. In our methodology we try to overcome some of the difficulties and address the limitations of our study in the discussion section. Collinearity is indeed a risk in the multi-variable analysis that we added to the methodology section.

ADJUSTMENTS:

Page 12: Multicollinearity between confounding variables and the telemedicine status is evaluated.

Page 16: DISCUSSION

The RELEASE-HF study is unique because of its observational study design, used to better understand heterogeneity in clinical effectiveness. To our knowledge, this is the first study using real-world data. The current design is chosen, because telemedicine is already implemented in various ways in health care settings in the Netherlands. Therefore, conducting a randomized controlled trial (RCT) does not fit the current care for HF patients. Additionally, using a national registry instead of a RCT allows us to observe all HF patients reducing selection bias that would otherwise be introduced by a RCT. A Dutch registry study previously showed that routinely collected data lead to a representative sample.[54] However, we have to be aware of confounding bias introduced by physicians or nurses who decide which patients may use telemedicine. We cannot completely avoid this bias, although we perform extensive confounding correction and conduct interviews with these health care professionals and asking about their local guidelines to understand the selection of patients in that hospital. Another limitation of registry-based studies is that the data to be collected are predetermined, which can lead to relevant missing variables. To overcome the missing data, such as telemedicine data on patient level or outcome measures, we aim to link other data sources to enrich the RELEASE-HF dataset.

13. Surprised not to see ventricular paced amongst the choice of heart rhythms. Units for NT-proBNP are usually ng/L or pg/mL. The criteria for iron deficiency are likely to change soon (see Ponikowski et al Eur Heart J August 2023).

Response: The reviewer rightfully points out that ventricular paced heart rhythm is missing from our data collection. As RELEASE-HF study, unfortunately we do not have the possibility to change the variable set of the HF registry of the NHR. Each registry within the NHR has a registration committee consisting of cardiologists who are mandated by their hospital. This committee determines standard sets of variables and periodically discuss the completeness and quality of data and patient-relevant outcomes. Probably in the HF committee the cardiologists decided not to include ventricular paced as choice of heart rhythms. The same goes for the units of NT-proBNP. The units are decided by the HF committee. We will convert the units (pmol/l) to the more common units as per the reviewers suggestion.

We are grateful that the reviewer notified us about the criteria for iron deficiency that are likely to change soon. We informed the NHR about these three comments, and they are considering incorporate the changes at the next national HF registry update.

14. There are many classes of diuretic of which loop diuretic is the most important for heart failure. It is critical that the authors gather and report data by diuretic class.

Response: Collecting data about the classes of diuretics is indeed important. Unfortunately, as explained in point 13, RELEASE-HF depends on the variables included in the HF registry. In RELEASE-HF we plan to link the data with other national registries, like PHARMO (i.e., national database for drug utilization) to include variables which are not included in the HF registry. In the main text we specify that we are planning to receive data on diuretics class via another national registry.

ADJUSTMENT:

Page 7: Variables of interests are visits to the outpatient clinic, to the emergency room, and to the general practitioner related to HF, admission days at an intensive care unit, and treatment characteristics specific to HF (i.e., medication as class of diuretics)."

VERSION 2 – REVIEW

REVIEWER	iyngkaran, pupalan Royal Darwin Hospital, Cardiology
REVIEW RETURNED	16-Oct-2023

GENERAL COMMENTS	Substantive improvements - i've never seen a rebuttal as good as this. no issues. NB// I am presuming the first article (pg 1 is track removed and the latter track changes.
---

REVIEWER	Cleland , J Glasgow University , School of Cardiovascular & Metabolic Health
REVIEW RETURNED	08-Nov-2023

GENERAL COMMENTS	The authors have responded well to my comments. The limitations of the design are now much clearer. However, I think even greater hesitation is required when suggesting that observational data can be used to show a treatment effect.
---

	Observational studies show the outcome with an intervention rather than the response to it (the reference below examines the problem with respect to CRT, where observational data have been so often misleading. It is perhaps worth referencing as a warning to the reader – caveat emptor! Cleland et al. Cardiac resynchronization therapy: are modern myths preventing appropriate use? J Am Coll Cardiol. 2009;53:608-611. PMID: 19215836 DOI: 10.1016/j.jacc.2008.10.040). Greater clarity on what is being analysed at an individual patient level and at a hospital level is required. This might be illustrated in diagrammatically. I am guessing this means that  A hospital declares whether they offered a patient received TM or not and whether this was i) accepted by the patients and, if so, ii) implemented The hospital generally provides information on their standard TM offer (can this change over time? It often will) It is assumed that if a patient receives TM they will all get the same offer from a particular hospital However, analysis of outcomes is done on an individual patient basis There are some spelling mistakes (eg: finial, strenghts) and grammatical errors (eg: split infinitives) that I assume will be sorted in copy-editing by the Journal. I think the word 'unique' should be deleted from the first line of the discussion. There are many observational studies of telemonitoring, although few at this scale. I am not a great fan of the term "real world data". Do patients exist in an unreal world. Terms such as 'representative of (daily) clinical practice might be more understandable and avoid jargon.
--	---

VERSION 2 – AUTHOR RESPONSE

Reviewer: 1

Dr. pupalan iyngkaran, Royal Darwin Hospital, Flinders University

Comments to the Author:

Substantive improvements - I've never seen a rebuttal as good as this. No issues.

Response:

We appreciate the compliment of the reviewer. Thank you for reviewing our manuscript.

Reviewer: 2

Dr. J Cleland , Glasgow University , University of Glasgow

Comments to the Author:

The authors have responded well to my comments

Response: We thank the reviewer for their positive appraisal of the rebuttal.

1. **The limitations of the design are now much clearer. However, I think even greater hesitation is required when suggesting that observational data can be used to show a treatment effect. Observational studies show the outcome with an intervention rather than the response to it (the reference below examines the problem with respect to CRT, where observational data have been so often misleading. It is perhaps worth referencing as a warning to the reader – caveat emptor! Cleland et al. Cardiac resynchronization therapy: are modern myths preventing appropriate use? J Am Coll Cardiol. 2009;53:608-611. PMID: 19215836 DOI: 10.1016/j.jacc.2008.10.040).**

Response: We are glad to read that adding the discussion paragraph to the manuscript has clarified some limitations in the chosen study design. We understand the reviewer's sense of concern about the interpretation readers could have when not fully aware about the type of outcome observational studies provide. We added a sentence about how to interpret the results of an observational study to the discussion paragraph and added the suggested reference.

ADJUSTMENT:

Page 16-17: DISCUSSION

*The RELEASE-HF study is a large-scale observational study to better understand heterogeneity in clinical effectiveness between patients using telemedicine. To our knowledge, this is the first study using routine clinical care data. The current design is chosen, because telemedicine is already implemented in various ways in health care settings in the Netherlands. Therefore, conducting a randomized controlled trial (RCT) does not fit the current care for HF patients. Additionally, using a national registry instead of a RCT allows us to observe all HF patients reducing selection bias that would otherwise be introduced by a RCT. A Dutch registry study previously showed that routinely collected data lead to a representative sample.[54] In addition, we have to be aware of confounding bias introduced by physicians or nurses who decide which patients may use telemedicine. We cannot completely avoid this bias, although we perform extensive confounding correction and conduct interviews with these health care professionals and asking about their local guidelines to understand the selection of patients in that hospital. **Consequently, it has been argued that results from observational studies describe a patient outcome when using the intervention rather than assessing the response to the intervention [55].** Another limitation of registry-based studies is that the data to be collected are predetermined, which can lead to relevant missing variables. To overcome the missing data, such as telemedicine data on patient level or outcome measures, we aim to link other data sources to enrich the RELEASE-HF dataset.*

2. **Greater clarity on what is being analysed at an individual patient level and at a hospital level is required. This might be illustrated in diagrammatically.**

I am guessing this means that:

- a) A hospital declares whether they offered a patient received TM or not and whether this was i) accepted by the patients and, if so, ii) implemented
- b) The hospital generally provides information on their standard TM offer (can this change over time? It often will)
- c) It is assumed that if a patient receives TM they will all get the same offer from a particular hospital
- d) However, analysis of outcomes is done on an individual patient basis

Response: We agree with the reviewer that greater clarity can be given in the manuscript about how the data collected at the hospital level and patient level contribute to the analysis on an individual patient level. Figure 2 illustrates which data source is used to collect data on the individual patient and which data source leads to data collected at the hospital level. Based on reviewer suggestions, we illustrated the data flow to show how the type of data collected on patient level and on hospital level lead to the analyses.

ADJUSTMENT:

Page 9: Figure 3 illustrates the flow of data collected at the hospital level and at the patient level.

Figure 3. This figure depicts how telemedicine characteristics are collected at the hospital level and at the individual patient level. It clarifies the source of the collected clinical content used for the subgroup analyses rather than the type of data source.

3. There are some spelling mistakes (eg: finial, strenghts) and grammatical errors (eg: split infinitives) that I assume will be sorted in copy-editing by the Journal.

Response: We appreciate the suggestion to correct spelling mistakes and grammatical errors in copy-editing by the Journal. We edited and improved the English language throughout the manuscript by an official translation service.

4. **I think the word ‘unique’ should be deleted from the first line of the discussion. There are many observational studies of telemonitoring, although few at this scale.**

Response: We thank the reviewer for pointing out that it is not the *observational study design* that makes the RELEASE-HF study unlike other studies about telemedicine, but rather the scale at which it is conducted.

ADJUSTMENT:

Page 16: The RELEASE-HF study is a large-scale observational study to better understand heterogeneity in clinical effectiveness between patients using telemedicine.

5. **I am not a great fan of the term “real world data”. Do patients exist in an unreal world. Terms such as ‘representative of (daily) clinical practice might be more understandable and avoid jargon.**

Response: We understand the reviewers’ opinion about using the term ‘real world data’. We agree that it is better to avoid jargon that triggers discussion.

ADJUSTMENTS:

Page 4:

A strength of this study is using nationwide routine clinical care data linked to national registry databases to capture current data on telemedicine use, heart failure patient characteristics, treatments, and clinically relevant outcomes with a one-year follow-up.

Page 11:

The Heart4Data consortium will establish possibilities for data linkage between the HF registry and other national registry databases, in collaboration with public partners (Figure 2).[28] At the moment of writing, a collaboration between these databases and the HF registry is not yet established for all external national registry databases.

Page 16:

Storage of linked datasets (HF registry with other national registry databases) will be part of the infrastructure established by the Heart4Data consortium.[28] RELEASE-HF study will follow these principles.

To our knowledge, this is the first study using routine clinical care data.

VERSION 3 – REVIEW

REVIEWER	Cleland , J Glasgow University , School of Cardiovascular & Metabolic Health
REVIEW RETURNED	02-Dec-2023
GENERAL COMMENTS	Well done - a few split infinitives in the grammar - but the copy editor should sort that (provided their grammar is up to scratch!)

VERSION 3 – AUTHOR RESPONSE